# The Effect of Cranial Cruciate Ligament Rupture on Range of Motion in Dogs

**DOI:** 10.3390/vetsci8070119

**Published:** 2021-06-24

**Authors:** Stefania Pinna, Francesco Lanzi, Chiara Tassani

**Affiliations:** Department of Veterinary Medical Sciences, University of Bologna, Via Tolara di Sopra 50, 40064 Ozzano dell’Emilia (BO), Italy; fralanz@gmail.com (F.L.); chiara.tassani2@unibo.it (C.T.)

**Keywords:** range of motion, cranial cruciate ligament, extension angle, flexion angle, dog

## Abstract

Range of motion (ROM) is a measure often reported as an indicator of joint functionality. Both the angle of extension and that of flexion were measured in 234 stifle joints of dogs with cranial cruciate ligament (CCL) rupture. The aims of this study were to investigate the correlation between CCL rupture and alterations in the range of stifle joint motion and to determine whether there was a prevalence modification of one of the two angles. All the extension and flexion angles were obtained from clinical records and were analysed in various combinations. A significant relationship was found between normal angles and abnormal angles; concerning the reduction in the ROM, a significant prevalence in the alteration extension angle was found. Of the 234 stifles, 33 (13.7%) were normal in both angles. These results could offer important insights regarding the influence of CCL rupture on compromising the ROM. This awareness could be a baseline for understanding the ability of surgical treatment to restore one angle rather than another angle, to address the choice of treatment and to help physiotherapists in their rehabilitation program.

## 1. Introduction

Range of motion (ROM) is the ability of a joint to move between positions to its maximum potential in relation to the three anatomic planes, the sagittal, frontal and transverse planes. Concerning the stifle joint, it is the movement in the sagittal plane, defined as either flexion or extension, which is usually studied [1]. The measurement of the joint angle is achieved using clinical goniometry, which gives an indication of the joint function and the eventual severity of the pathology [1,2].

Goniometry has been validated in humans [3,4,5,6], and several methods of measurements have been documented and compared [7,8,9]. In veterinary medicine, goniometry has been studied in different species, such as cats [10], horses [11,12,13] and sheep [14]. The reliability of goniometry in dogs was documented by Jaegger in 2002, comparing the measurements made with radiography with the measurements made by a transparent plastic goniometer in 16 healthy Labrador Retrievers [2].

Physiological and/or pathological causes can modify the ROM. Variations can be based on activity and the breed of dog; the body’s condition and joint health can affect the ROM as a result of changes in the shapes of the joints, capsules, periarticular tendons, muscles and ligaments [1,15,16,17]. Therefore, a lack of joint motion can occur following injury or surgical treatment or in patients with chronic diseases, such as osteoarthritis (OA) or muscle contracture. During the OA process, osteophytes, capsular adhesions and shortened ligaments are generated and lead to a mechanical reduction of the ROM. Another cause of ROM abnormality could be pain, as maximum joint motions lead to a stretching of the joint capsule and, consequently, increases the nociceptor activity [18,19]. Changes in the ROM can be due to reducing the extension angle or increasing the flexion angle or both alterations. The result is obviously a reduction in the ROM.

Reducing the ROM could lead to lameness, which is greater when the loss of ROM is greater. The incomplete extension or flexion of a joint is associated with some functional disabilities, such as jumping into a car, rising from a recumbent position or climbing stairs [19].

Goniometry has been used in canine orthopaedics [20] to assess the treatment efficacy of diseases involving shoulders [21], elbows [19,22,23], stifles [19,24,25,26,27,28,29] and hip joints [30,31]. The most common injury involving the canine stifle joint is cranial cruciate ligament (CCL) rupture [32,33]. The incorrect use of the affected limb, i.e., due to pain, chronic disease, joint involved or OA, leads to muscle atrophy and a reduced ROM [32]. Several studies have used the measurements of the ROM in dogs with a CCL rupture to evaluate the functional outcome after surgical repair [32,33,34,35,36,37,38,39,40,41] or postoperative rehabilitation. It is currently known that physiotherapy is useful for improving the functioning of a restricted joint [24,42,43].

The aim of the present study was to investigate the relationship between a CCL rupture at the time of diagnosis and the changes in stifle joint motion using the values of the extension and flexion angles. The hypotheses were that there was a correlation between the two angles and the loss of extension was more frequent than the alteration of flexion in dogs with a CCL rupture.

## 2. Materials and Methods

The Bologna Healing Stifle Injury Index—Clinical Record (BHSII—CR) of dogs with a spontaneous CCL rupture was obtained from the archives of the University Hospital of the Department of Veterinary Medical Sciences, University of Bologna, Ozzano dell’Emilia (Bologna), Italy from 2011 to 2018, and they were retrospectively reviewed. Of the various items in the index, the assessment of both extension and flexion movements was retrieved. The diagnosis of a CCL rupture was made at the time of the compilation of the BHSII using a complete orthopaedic examination.

All items of the BHSII concern a damaged stifle joint; the assessment method selected was the Likert scale, which has five levels of answers (scores from 0 to 4) [44]. Based on the median values published by Jaegger [2], such as extension angle = 162° and flexion angle = 41°, which referred to the stifle joints of healthy dogs, the authors of the BHSII [44] created the following scale of values. The extension angle (EA) was set as scores 0 = 162–158°, 1 = 157–153°, 2 = 152–148°, 3 = 147–143° and 4 = 142–138° and the flexion angle (FA) as scores 0 = 41–45°, 1 = 46–50°, 2 = 51–55°, 3 = 56–60° and 4 = 61–65° (see https://www.frontiersin.org/articles/10.3389/fvets.2019.00065/full#supplementary-material Accessed on 1 April 2021) [44]. These scores were indicative of no alteration (score 0) or a slight, mild, moderate or severe reduction in the degree of extension and/or increase in flexion (scores 1, 2, 3 and 4, respectively). Each angle of extension and flexion was assessed individually.

At the time of the BHSII-CR compilation, all the dogs were sedated and placed in lateral recumbency. Sedation was required to obtain adequate relaxation of the patient in order to perform specific measurements and manualities, i.e., angle of motion and drawer movement, without interference due to abnormalities, namely pain or muscle contracture. The same protocol was used for each patient. All the dogs were sedated with dexmedetomidine (Dextroquillan; ATI, Bologna, Italy) 3–5 mcg/kg or medetomidine (Sedastart; Ecupharma s.r.l, Milano, Italy) 5–10 mcg/kg intramuscularly (IM) in combination with butorphanol (Nargesic; Acme S.r.l., Reggio Emilia, Italy) 0.1–0.2 mg/kg. The dexmedetomidine or medetomidine were administered at the anaesthetist’s discretion. The range of joint motion was measured as the degree of extension and flexion using a plastic goniometer. As described by Jaegger, the arms of the goniometer were aligned with the tibia shaft and with the longitudinal axis of the femur, which were the line that joins the lateral femur epicondyle and the greater trochanter, respectively [2]. Measurements of the extension and flexion angles were reported in the BHSII-CR of each dog. The measurements were taken three times by the veterinarians of the orthopaedics staff of the University Hospital and were averaged [33,36,45], and the corresponding score was then selected on the BHSII-CR.

The inclusion criteria were a BHSI-CR completely filled out without limit by age, sex, breed, acute or chronic disease, duration of lameness, partial or complete CCL tears or signs of OA. If the BHSH-CR was missing information regarding the ROM, and other items of the clinical record, the dogs were excluded. The dogs with a BHSII that met the inclusion criteria were included in the injury group (IG), meaning those with a spontaneous CCL rupture.

### 2.1. Collection Data

All demographic data and the scores of stifle motion of each dog were collected in a spreadsheet (Office Excel 2019, Microsoft Corporation, Redmond, WA 98052-6399 United States). The groups of combination scores were classified as follows:NG = the normal group that included dogs having a score of 0 in both extension and flexion;EFAG = the extension and flexion abnormal group that included dogs having scores 1–4 in both extension and flexion;EAAG = the extension angle (reduced) abnormal group that included dogs with stifles having an extension angle with scores 1 to 4 and a flexion angle having a score of 0;FAAG = the flexion angle (increased) abnormal group that included dogs with stifles having an extension angle with a score of 0 and a flexion angle with scores 1–4.

The control group (CG) included twenty healthy dogs without a history or clinical signs of orthopaedic diseases in order to verify the reliability of the scores.

Based on body weight, the dogs in the IG were divided into three groups: Group A < 20 kg, Group B from 21 to 30 kg and Group C > 31 kg.

The ROM in the IG was assessed on the basis of the duration of the lameness and/or the time passed from the trauma, if known. An acute group (AG) and a chronic group (ChG) were created. The ROM in the IG was also assessed on the basis of muscle mass, and the normal muscle group (NMG) and the decreased muscle group (DMG) were obtained.

### 2.2. Statistical Analysis

The continuous data were evaluated using the Kolmogorov–Smirnov test for normal distribution; if they were rejected, nonparametric tests were carried out. The data were reported as the median, range (minimum and maximum values) and 95% confidence interval (CI). The categorical or discrete variables were evaluated as frequencies and/or percentages and processed using the chi-square test for trends. Spearman’s rank correlation coefficient was calculated to measure the relationship between the angles of extension and flexion. All the data were analysed using a statistical software program (MedCalcR Software 16.8.4, Ostend, Belgium). Significance for all the analyses was set at *p* < 0.05.

## 3. Results

### 3.1. Descriptive Analysis

A total of 234 dogs with a CCL rupture met the inclusion criteria (IG)). The age and body weight were reported as the median, range and 95% CI. All the demographic data are reported in Table 1.

The most common breeds included 67 crossbreed dogs, 28 Labrador Retrievers, 18 Boxers, 11 German Shepherds, 11 Rottweilers, 9 Maremma Sheepdogs, 8 American Staffordshire Terriers, 8 Cane Corso, 6 Beagles, 6 Yorkshire Terries, 4 Epagneul Bretons, 4 Jack Russell Terriers, 4 Maltesi, 3 Australian Shepherds, 3 Barboncini, 3 Bernese Mountain dogs, 3 Border Collies, 3 Doberman Pinchers and 3 Volpino Italiano. There were also two dogs of each of the following breeds: Bolognese, Bullmastiff, Chow Chow, Golden Retriever, Leonberger, Pyrenean Mastiff and Tosa Inu. There was also one dog each for the following breeds: Akita Inu, American Pit Bull Terrier, Bracco Italiano, Caucasian Shepherd, Cavalier King Charles Spaniel, Dogue de Bordeaux, English Setter, Fox Terrier, Great Dane, Lagotto Romagnolo, Miniature Pinscher, Pyrenean Shepherd, Riesenschnauzer, Saint Bernard, Samoyed, Siberian Husky, Toy Poodle and West Highland Terrier.

In the IG, the Mann–Whitney test was applied to evaluate the differences in age and/or weight between the male and female groups. Only the weight distribution was statistically significant (*p* = 0.0009).

The CG included six Golden Retrievers; three crossbreeds; three Border Collies; two Dachshunds and one dog each of the following breeds: Basenji, Belgian Malinois, Labrador Retriever, Lagotto Romagnolo, Newfoundland and Weimaraner.

### 3.2. Statistical Analysis


In the IG, the Spearman’s rank correlation coefficient revealed a significant level of correlation (*p* = 0.0031; rho = 0.192) between the extension and flexion angles.In the IG, each injured stifle angle of each dog was investigated. The scores of motion in both extension and flexion were then assessed as a normal angle (NA = score 0) or an abnormal angle (AA = scores 1, 2, 3 or 4). The data are reported in Table 2.In the IG, the chi-square test revealed a statistically significant relationship of the NAs and AAs between the two angle motion groups (EA and FA) (*p* = 0.01). There was a reduced ROM in 86.3% of the cases, and there was a significant prevalence in the alteration of the extension angle. The combinations of the scores were analysed: 13.7% of the dogs had a normal ROM, 50.9% of stifles had a reduced ROM due to an alteration of both movements, 19.2% had abnormal extension with normal flexion and 16.2% had abnormal flexion and normal extension (Figure 1 and Table 3). There was a prevalence of abnormal extension, having a contingency coefficient of 0.16.All five scores (0 to 4) were then evaluated to obtain more details. In the IG, the combinations of the scores of the EAs and FAs were statistically significant, with *p* = 0.0248 (Figure 2). Only 13.7% (n.33) of the stifles were normal. The stifles with normal EAs (score 0) were associated with abnormal FAs distributed in the slight prevalence score. The stifles with normal FAs (score 0) were associated with abnormal EAs mainly distributed in the slight and the mild scores. Overall, the alterations of the EAs were found in all the scores of severity; instead, the prevalence of the frequency of the alterations of the FA was found in score 1 (slight). The contingency coefficient was 0.33.The dogs were grouped based on weight: Group A < 20 kg, Group B from 21 to 30 kg and Group C > 31 kg. The demographic data are listed in Table 1. The chi-square test for the trends revealed a statistically significant difference in the EA among the dogs in Groups B and C, having greater alterations (*p* = 0.0098 and *p* = 0.0124, groups B and C, respectively). The alterations in the FA were not significant.No significant differences were found between males and females in either the extension or the flexion angles.All ruptures were present for more than 2 weeks. Dogs with known lameness and/or trauma occurrence within the past 30 days were classified as the acute group (AG). There were 16 dogs (6.84%) (out of 234) with acute injuries. The chi-square test for the relationship of NAs and AAs was not significant (*p* = 0.36). There were 218 (93.16%) dogs with chronic disease (Chronic Group: ChG), and the chi-square test revealed a significant difference in the normal and abnormal angles of motion (NAs and AAs), *p* = 0.027. In the ChG, the alteration of extension was prevalent at flexion, having a percentage of 19.3%, which was the same as the value found in the IG (Table 3).The IG was divided into two groups of normal and abnormal muscle mass. The investigation revealed that NMG included 22 dogs, and there was not a significant difference between normal and abnormal angles (*p* = 0.55). The DMG consisted of 212 dogs, and there was a significative difference between NAs and AAs (*p* = 0.019). The percentage of the combination of normal and abnormal extension and flexion had the same trend of IG (Table 3).In the CG, 20 healthy dogs showed 85% of the dogs having a normal ROM. The remaining stifles (15%) had scores slight or mild; the latter dogs weighed > 20 kg and were >7 years of age (Figure 1 and Table 2 and Table 3).


## 4. Discussion

The aim of the present study was to retrospectively investigate a correlation between a CCL rupture and the alterations in the ROM in stifle joints. The analysis of 234 stifle joints revealed that the CCL rupture led to a reduced ROM, likely due to a modification of the joint biomechanics and/or modification of the periarticular structures [32]. The authors’ hypotheses concerning a correlation between the alterations of the two angles of motion and the fact that the alterations in the extension angle were statistically significant and greater than those in the flexion angle were confirmed. A previous study [46] carried out on healthy dogs reported that the movement across the joint is flexor from ground contact until mid-stance and then extensor until the end of the stance phase. In a dog with a CCL rupture in which the extension is limited, its gait could also be compromised, and it is assumed that, as in human knees, abnormal extension is less tolerable than abnormal flexion [36,47]. The above-mentioned considerations provided the purpose of the investigation in the present study, as, to the best of the authors’ knowledge, no previous reports have examined the incidence and prevalence in movement changes in dogs as a consequence of a CCL rupture.

Not investigating the relationship of abnormal ROM with other parameters, such as the severity or duration of lameness and scores of OA, was a limitation and potential flaw or shortcoming of this study. It is known that the duration of lameness could be a cause of OA progression [48] and that OA could lead to stiffness, a mechanical block from osteophytes [49,50] and a loss of ROM [18,19]. Furthermore, in the literature, the lack of a relationship between the functional capacity of dogs with OA and radiographic evidence of disease has been described [51]. The aforementioned considerations were addressed in the present study in order to verify whether a CCL rupture could be responsible for the modification of one of the two angles of motion, or both, by means of evaluating a large number of cases without limits of recruitment. Another limitation could be the breed itself, keeping in mind that goniometric measurements of the stifle joints could be different based on different breeds of healthy dogs [1,15,16,17]; however, in the present study, the most common breed was the crossbreed.

The present study assessed the angles using a Likert scale from 0 to 4 in place of the exact degree of each angle, obtained from the BHSII. The EA and FA measurements were taken three times and averaged [33,36,45], and the value obtained was put into a range of one of the five scores. Each score had a range of five degrees, so the authors assumed that intra-observer and interobserver variability could be avoided. As the range of each scale is known, anyone can deduce the degree of the angle in order to compare it with other studies.

As expected, the authors found a correlation between the angles of extension and flexion, as they are movements of the same joint. Despite this, there was a percentage prevalence of the extension angle at the time of the diagnosis of CCL ruptures in dogs weighing > 21 kg. This prevalence was in contrast with Sabanci’s study in which the body weight of healthy dogs was the most influential factor regarding the flexion angle. It is likely that this difference could have been due to the joint disease [16].

An investigation regarding the length of time of the disease found a difference in sample sizes between the acute and the chronic groups, (16 vs. 218, respectively), which did not allow the application of correct statistical tests. However, the results obtained from the chi-square test assumed that a longer duration of the lesion could affect the alteration of the ROM; the prevalence in the frequency of extension alteration was similar in the entire IG.

Muscle mass has an influence on the ROM; in particular, an atrophied muscle can modify the flexion angle. In other words, it can produce a reduction in the flexion degree. This consideration was in contrast with the results of the present study, in which the alteration of the extension was prevalent in the flexion angle. Therefore, the authors assumed that the CCL rupture led to an alteration of extension angle due to other regions.

Several studies have investigated variations in ROM based on pain and muscle mass [32,33,34,35,36,37,38,39,40,41]. Knowledge of the causes of a reduced flexion angle is associated with muscle disuse and atrophy, while causes of an increased flexion angle are associated with changes in the joint structures. Concerning extension, joint alterations can be acute (i.e., capsular effusion) [50,52,53], which, if treated quickly, could favour the recovery of normal extension. When the alterations are chronic, they have already developed OA [50,52,53] and are responsible for biomechanical influences that could result in a greater difficulty of being resolved and in the consequent persistence of the reduction of the extension angle. Pain is always a contributing factor to alterations in both angles [18,19]. All these variables can be concomitant and consequent to the CCL rupture; therefore, the authors assumed only the certain biomechanics and degenerative alterations that occur in the stifle joints with a CCL rupture as the main parameter of ROM modification, reporting a prevalence of reduced extension. Despite this, it was found that 13.7% (n. 33) of the stifles had no change in ROM in contrast to what was expected. Additional studies carried out on a greater number of stifles are necessary to investigate the possible maintenance of normal angles of motion in dogs with a CCL rupture. Concerning the evaluation of the correlation between the ROM and OA, the authors consider this report to be a preliminary study. Once the hypothesis is confirmed, it will be possible to carry out future studies to investigate the influence of OA on the variations in the angles of extension and/or flexion.

As expected, the control group was helpful in showing that in the absence of cruciate rupture ROM may be normal, except for older or overweight animals, which can be assumed to have other influencing factors, e.g., OA.

In the present study, all the measurements were taken with the dogs sedated in order to eliminate the interference of pain or tendon, ligament, capsule and muscle contractures. In diseased joints, the ROM may be restricted in response to a painful sensation in awake patients [2]. In effect, in Jaegger’s study, the measurements of the ROM in dogs awake and sedated were identical [2]. However, the authors pointed out that their study regarded a healthy population, and these results would be unlikely in dogs with degenerative joints [2]. For the same reason, in 2010, Mostafa [38] also considered the ROM for all joints in dogs under sedation in order to minimise the possible restriction of a painful response. The present study confirmed the results of Mostafa’s study [38], in which Labradors with and without CCL deficiency were compared and the goniometric measurements of extension and flexion were abnormal and statistically significant for the extension, as in the present study.

The results of this study were discordant with the Jandy report [36], which assessed 412 dogs with a CCL rupture that had both the angles of extension and flexion as normal [2]. These normal values were reported prior to surgical treatment in several dogs and, also, after tibial plateau levelling osteotomy (TPLO) treatment, and there was no significant correlation between flexion and extension in the dogs examined. Moreover, a loss of flexion or extension greater than 10° was associated with greater lameness [36]. Other studies have reported alterations in the ROM after surgical treatment and have often only stated whether or not there was an improvement, without reporting the degree of the angle measurements [32,33,35,37,38,39,41]. In 2010, Au reported the pre- and postoperative ROM as not being significant [36]; however, he did not report the degree of the angles. The Moeller and Mölsä studies reported the angles measured after TPLO treatment and compared them to the contralateral limbs [32,33]; however, they did not indicate the degrees of the angles prior to surgery. In 2013, McDonald reported that the ROM in an affected limb was significantly less than that of a healthy contralateral limb [41]; however, he did not compare the flexion and extension, as was done in the present study. The papers in the literature did not use the same criteria for evaluating the ROM; therefore, the results of the above-mentioned reports could not be compared with those of the present study, which reported degree scores for both angles.

Given that, for the most part, gait is performed with the limb in weight-bearing and in the stance phase of extension, this information and that mentioned above could provide input for a screening program and preventive rehabilitation. Additional studies are important for understanding each of the most frequently used surgical techniques that can compromise and/or favour one of the angles of motion or both in order to address the choice of treatment.

## 5. Conclusions

In conclusion, the authors believe that the results of this study could offer important information regarding the influence of a CCL rupture in compromising the ROM. This awareness could be a baseline for additional studies regarding the ability of surgical treatments to restore one angle rather than another. Based on these findings, physiotherapy could also be helpful with a novel approach in a nonsurgical and/or postsurgical treatment program or in preparing dogs prior to surgery.

## Figures and Tables

**Figure 1 vetsci-08-00119-f001:**
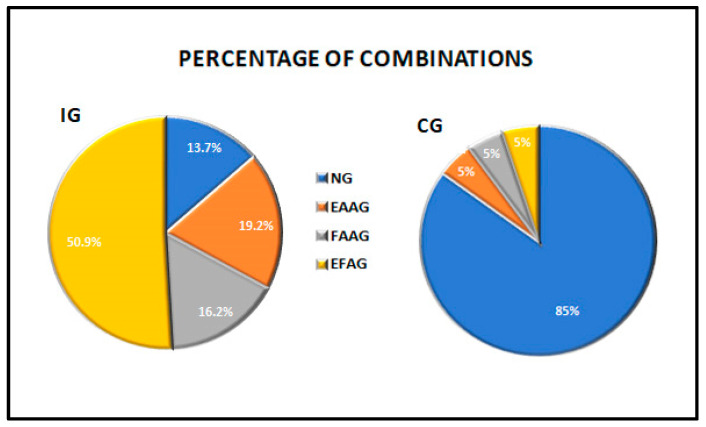
Pie chart showing the distribution of the combinations of the normal and/or abnormal angles of extension and flexion in the IG and the CG. IG: injury group; CG: control group; NG: normal group; EAAG: extension angle abnormal (reduced) group; FAAG: flexion angle abnormal (increased) group; EFAG: extension and flexion angle abnormal group.

**Figure 2 vetsci-08-00119-f002:**
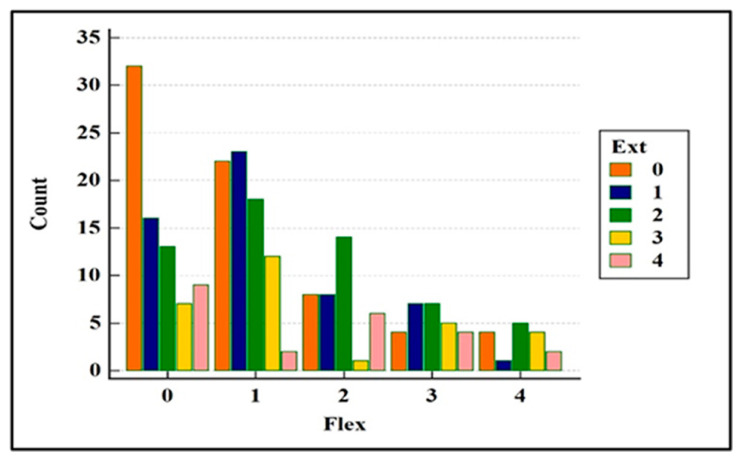
The chart shows the scores (0–4) of the flexion angle in combination with extension angle. Count: number of stifles.

**Table 1 vetsci-08-00119-t001:** Descriptive characteristics of the dogs. Median, range (minimum and maximum values) and 95% CI for the medians are reported for age (y) and weight (kg).

Dog/Stifle	n.	%	Age			Weight		
			Median	Range	95% CI	Median	Range	95% CI
IG	234	100	6	1–13	6–7	30	4–75	28–32
Male	99	42.3	6	1–13	5–7	35 *	4–75	30–38
Female	135	57.7	7	1–13	6–7	28	4–56	26–30
Group A	64	27.4	8	2–13	7–9.3	10	4–18	8–11
Group B	91	38.9	6	1–12	5–6.3	29	21–35	28–30
Group C	79	33.8	5	1–13	5–6	42	36–75	40–44.9
CG	20	100	6	1–12	3–7.8	28	14–55	22–29.8

IG, Injury Group; Group A, <20 kg; Group B, 21 to 30 kg; Group C, >31kg; CG, Control Group; * significant *p*-value.

**Table 2 vetsci-08-00119-t002:** Distribution in frequency (n.) and percentage (%) of the extension and flexion angle scores in both the CCL injury group and the control group.

Angle Motion		Scores					Total AA
		0 (normal)	1 (slight)	2 (mild)	3 (moderate)	4 (severe)	
IG	EA	70 (30%)	54 (23%)	58 (25%)	29 (12%)	23(10%)	164 (70%)
	FA	77 (33%)	76 (32%)	38 (16%)	27 (12%)	16 (7%)	157 (67%)
CG	EA	17 (85%)	2 (10%)	1 (5%)	-	-	3 (15%)
	FA	17 (85%)	3 (15%)	-	-	-	3 (15%)

IG: Injury Group; CG: Control Group; EA: extension angle; FA: flexion angle; AA: abnormal angle (scores 1, 2, 3 or 4).

**Table 3 vetsci-08-00119-t003:** Percentage of score combinations of the normal and abnormal angles in the various groups.

	IG	CG	AG	ChG	NMG	DMG
NG (0 + 0)	32 (13.7%)	17 (85%)	4 (25%)	28 (12.8)	4 (18.2%)	28 (13.2%)
EAAG (1 + 0)	45 (19.2)	1 (5%)	3 (18.8%)	42 (19.3%)	5 (22.7%)	40 (18.9%)
FAAG (0 + 1)	38 (16.2%)	1 (5%)	2 (12.5%)	36 (16.5%)	3 (13.6%)	35 (16.5%)
EFAG (1 + 1)	119 (50.9%)	1 (5%)	7 (43.8%)	112 (51.4%)	10 (45.5%)	109 (51.4%)
Total dog/group	234	20	16	218	22	212

IG: injury group; CG: control group; AG: acute group; ChG: chronic group; NMG: normal muscle group; DMG: decrease muscle group; NG: normal group; EAAG: extension angle abnormal (reduced) group; FAAG: flexion angle abnormal (increased) group; EFAG: extension and flexion angle abnormal group.

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
