# Peer review of "The Effect of Cranial Cruciate Ligament Rupture on Range of Motion in Dogs"

_vetsci, 2021, doi:10.3390/vetsci8070119_

Round 1
Reviewer 1 Report
This would be an interesting study in principle, but unfortunately I have massive problems, which I describe in detail below.
ROM cannot be measured only in extension and flexion. This should at least be mentioned.
L. 45: A changed ROM can be correlated to the degree of lameness, but that it is the reason for lameness cannot be said so.
L. 51: That the lameness itself leads to an altered ROM may be possible, but the main reason is surely the joint involved (pain etc.).
L. 58: I am unclear what is meant by the hypothesis that there is a correlation between the angles? Could this be made more concrete? For example, that the loss of flexion is greater when the restriction in extension is greater?
L. 62 ff: Here it is not quite clear to me whether the same dogs were used for the present study as in the cited study, or whether the data from existing medical records were used (this is how I read it from line 63). If the former is correct: please mention, if the latter is correct: in which time period were the data collected?
L 74 ff: even if this data has already been published, it would make it much easier to understand the study if this was better explained. So what would it mean, for example, if extension = 4 but inflection = 0? I assume that each angle is assessed on its own? You can mention that briefly, then it becomes easier to understand.
L 94: in principle it is clear that the angle of flexion increases, i.e. has higher values. I just wonder whether readers who are not so well read into the field are not confused by this. Wouldn't it be better to explain at the beginning how the angles are measured (with an illustration, for example) to explain this?
L 96: Control group: was this also studied under sedation? Or does this again refer to a published study? What conclusions are drawn from the values measured in this group? Here, too, some of the animals had altered values. Which animals (breed etc.) were used here?
In general: there is no information on clinical aspects. Changes in ROM depend on many factors. For example, whether muscle atrophy is already present, how long the disease has existed. No distinction is made between animals with partial or complete ruptures. This could influence the measurements. It would also have been of enormous importance to determine which dogs were overweight. Any thick layer of fat will reduce flexion.
L111: does TG also mean the healthy dogs, or only those with cruciate ligament tears?
L 131: all dogs (healthy and sick) or only sick dogs?
L134: scores slight or mild: this refers to extension and/or flexion, here I come to my comment above - without explanation this is not understandable and it should not just be succinctly referred to another study. If this refers to the individual angles, why is there no table for the CG group?
143: Maybe it is my lack of statistical understanding, but when first in l. 143 it says: "here was a significant prevalence in the alteration of the extension angle" and then in l. 147 "here was a lesser prevalence of abnormal extension", this seems to contradict each other. Can this be explained better?
L 116 ff and 165 ff: I have concerns about the different breeds in the study, differences in ROM cannot be mitigated by weight alone. Muscle mass alone is a decisive factor. And this is where breeds of the same body mass (or groups) differ. For example, a male Labrador may well weigh more than 31 kg, but a comparison with a bullmastiff is probably not permissible, especially since, to my knowledge, there are no studies comparing such different breeds. Even if this is touched upon in the discussion - why was not at least the muscle circumference, and as described above overweight evaluated. Or try to form a group from the existing data where the body shape is at least similar and look at the certainly existing data on surgical findings (partial vs. complete rupture). A study with fewer animals that is better analysed would provide valuable information, but for me there are too many influencing factors.
Author Response
ROM cannot be measured only in extension and flexion. This should at least be mentioned.
Response 1:Thank you for your suggestion. The ROM definition has been changed as recommended. See the Introduction section.
L.45: A changed ROM can be correlated to the degree of lameness, but that it is the reason for lameness cannot be said so.
Response 2:Thank you for your comment. The sentence has been changed. The main causes of ROM reduction (chronic disease, joint involved, pain, etc.) have been listed in the paragraph preceding the line you refer to.
L.51: That the lameness itself leads to an altered ROM may be possible, but the main reason is surely the joint involved (pain etc.).
Response 2: I agree with your comment. Perhaps your concern regarding lameness comes from the phrase "the incorrect use of the affected limb ... "line 55”. There are several causes which lead to "incorrect use ... "as described in the previous answer. A brief addition has been made.
L.58: I am unclear what is meant by the hypothesis that there is a correlation between the angles? Could this be made more concrete? For example, that the loss of flexion is greater when the restriction in extension is greater?. .
Response 3: The sentence has been modified. Restriction in extension is more frequent, and not greater, than loss of flexion.
L.62 ff: Here it is not quite clear to me whether the same dogs were used for the present study as in the cited study, or whether the data from existing medical records were used (this is how I read it from line 63). If the former is correct: please mention, if the latter is correct: in which time period were the data collected?
Response 4: Yes, the latter is correct. The time of collection data was 7 years. As reported by Pinna 2019: “the BHSII was developed in various steps between 2006 and 2010, following Brown’s suggestions, and the data were collected at the Department of Veterinary Medical Sciences, University of Bologna, IT, from 2011 to 2018.” The time of collection has been added to the Materials and Methods section.
L 74 ff: even if this data has already been published, it would make it much easier to understand the study if this was better explained. So what would it mean, for example, if extension = 4 but inflection = 0? I assume that each angle is assessed on its own? You can mention that briefly, then it becomes easier to understand.
Response 5: Yes, each angle was assessed on its own. The significance of the scores has been defined on line 81: “The scores of each angle (from 0 to 4) were indicative of no alteration, or slight, mild, moderate or severe alteration of joint movements, respectivelly.
L 94: in principle it is clear that the angle of flexion increases, i.e. has higher values. I just wonder whether readers who are not so well read into the field are not confused by this. Wouldn't it be better to explain at the beginning how the angles are measured (with an illustration, for example) to explain this?
Response 6: I have followed your suggestion. The angles were measured with a plastic goniometer as Jaegger reported. The description of how the angles were measured has been added to the Materials and Methods section.
L 96: Control group: was this also studied under sedation? Or does this again refer to a published study? What conclusions are drawn from the values measured in this group? Here, too, some of the animals had altered values. Which animals (breed etc.) were used here?
Response 7: The control group data also derived from the study published in 2019. The healthy dogs were sedated for investigations other than orthopaedic. The list of breeds has been added to the Results section, and the distribution of frequency in the scores in both angles has been added to Table 2. Finally, the percentages of the score combinations have been reported in a new table (Table 3). A comment regarding the results has been added to the Discussion section.
In general: there is no information on clinical aspects. Changes in ROM depend on many factors. For example, whether muscle atrophy is already present, how long the disease has existed. No distinction is made between animals with partial or complete ruptures. This could influence the measurements. It would also have been of enormous importance to determine which dogs were overweight. Any thick layer of fat will reduce flexion.
Response 8: Thank you for your suggestion. I have added an assessment of the acute (AG) and chronic (ChG) groups, AG <30 days and ChG > 30 days from the start of the injury, respectively, identified in the clinical records using the time of lameness and/or reported trauma. An investigation regarding the influence of these two groups on ROM has been added to the manuscript.
Only three dogs had a partial rupture, and no statistical survey group was utilised.
Furthermore, the effect of muscle mass on ROM has been calculated and a statistical analysis of the groups with normal (NMG) or decreased muscle mass (DMG) has been carried out and added to the study. Overweight was not reported. Some clinical signs were not investigated and were explained as study limitations in the Discussion section.
L111: does TG also mean the healthy dogs, or only those with cruciate ligament tears?
Response 9: TG means Total Group, only the dogs with cranial cruciate ligament tears. Actually, the acronym is misleading, so TG has been changed to IG (Injury Group).
L 131: all dogs (healthy and sick) or only sick dogs?
Response 10: Following the suggestions and questions of the reviewers, the statistical analysis has been extended to the healthy and sick groups, and to the other newly created groups: acute / chronic, normal or altered muscle mass.
L134: scores slight or mild: this refers to extension and/or flexion, here I come to my comment above - without explanation this is not understandable and it should not just be succinctly referred to another study. If this refers to the individual angles, why is there no table for the CG group?
Response 11: Thank you for your suggestion. I tried to improve the text, and the measurements with the goniometer have been detailed. The results obtained for the GCs have been added to Tables 2 and 3.
143: Maybe it is my lack of statistical understanding, but when first in l. 143 it says: "here was a significant prevalence in the alteration of the extension angle" and then in l. 147 "here was a lesser prevalence of abnormal extension", this seems to contradict each other. Can this be explained better?
Response 12: Thank you for your observation. I have deleted “lesser”.
L 116 ff and 165 ff: I have concerns about the different breeds in the study, differences in ROM cannot be mitigated by weight alone. Muscle mass alone is a decisive factor. And this is where breeds of the same body mass (or groups) differ. For example, a male Labrador may well weigh more than 31 kg, but a comparison with a bullmastiff is probably not permissible, especially since, to my knowledge, there are no studies comparing such different breeds. Even if this is touched upon in the discussion - why was not at least the muscle circumference, and as described above overweight evaluated. Or try to form a group from the existing data where the body shape is at least similar and look at the certainly existing data on surgical findings (partial vs. complete rupture). A study with fewer animals that is better analysed would provide valuable information, but for me there are too many influencing factors.
Response 13: Thank you for your suggestion. Some studies have been published regarding ROM in various breeds (see references 1, 15, 16, 17 and 19). The most interesting was Sabanci’s report which investigated ROM in seven breeds of healthy dogs. In the present study, I assessed dogs with CCL rupture which was the primary aim. The impact of muscle mass on ROM has been investigated, and a statistical analysis of groups with normal or decreased muscle mass was carried out and added to the study. Interesting considerations were raised. See the Results and Discussion sections.
No evaluations for breeds were carried out due to the great differences in the size of the samples.
Reviewer 2 Report
Dear Authors,
- It´s important to write the protocol of sedation, that all dogs received. Because some drugs give better muscle relaxation and this will interfere with measurements.
- How many researchers have taken the measurements of the joints, or was all the time the same researcher?
Author Response
- It´s important to write the protocol of sedation, that all dogs received. Because some drugs give better muscle relaxation and this will interfere with measurements.
Response 1: Thank you for your observation. The protocol of sedation has been added to the Materials and Methods section.
- How many researchers have taken the measurements of the joints, or was all the time the same researcher?
Response 2: Thank you for your suggestion. I agree that it is an important point to know. The measurements were taken by the veterinarians of the orthopaedics staff of the University Hospital. Details have been added to the Materials and Methods section.
Reviewer 3 Report
In general, this scientific article is very carefully written and documented with advanced statistical research methods. However, during the reviewing several questions have been raised. Several important data are missing:
- What kind of CCL rapture was observed in experimental dogs (acute rapture, chronic rapture or partial tears?)
- What was a method of CCL diagnosis ? Only physical exam or radiographs (x-rays) were also taken
- The duration of the lameness should be presented in table 1
- What was the period between the first clinical signs of CCL rapture and the full diagnosis.
- It is not known whether the arthritis of stifle joint was associated with the course of CCL rupture
Moreover, the conclusion does not constitute serious discovery but rather kind of a wishful thinking. In my opinion this study should be supported by clinical (surgical) trials.
Author Response
- What kind of CCL rapture was observed in experimental dogs (acute rapture, chronic rapture or partial tears?)
Response 1: Thank you for your useful observation. All the ruptures had been present for more than 2 weeks. I added an assessment of the acute and chronic groups, <30 days and > 30 days from the start of the injury, respectively, identified in the clinical records using the time of lameness and/or reported trauma. An investigation regarding the influence of these two groups on ROM has been added to the manuscript. Concerning the total or partial tears, investigation regarding the effect on the modification of angles of motion was not carried out because only three dogs had partial tears.
- What was a method of CCL diagnosis ? Only physical exam or radiographs (x-rays) were also taken.
Response 2: The diagnosis was made at the time of the compilation of the BHSII, using a complete orthopaedic examination. A sentence has been added to the Materials and Methods section.
Radiographic examination has been carried out to obtain more information on the presence of osteoarthritis and to carry out preoperative planning. As highlighted in the manuscript, no correlation was made with degrees of OA, as the main objective was to evaluate the existence of ROM variations in dogs with CCL rupture. Future studies will investigate this correlation.
- The duration of the lameness should be presented in table 1
Response 3: Unfortunately, in the medical records the duration of lameness was not always specified; therefore, I could not add the data (median, range for age and weight) to the Table 1. However, I was able to create two groups for duration <30 and> 30 days (acute and chronic injury). A statistical survey has been carried out on these groups.
- What was the period between the first clinical signs of CCL rapture and the full diagnosis.
Response 4: I understand that what you are asking me is lacking, however, many dogs come to the University Hospital on reference. In the anamnesis, this information was often missing. See answer no. 3.
- It is not known whether the arthritis of stifle joint was associated with the course of CCL rupture.
Response 5: Yes, it is. As I answered question in n. 2, x-rays were taken. I know this is a very important evaluation, but the aim of this study was to see how ROM changes in dogs with CCL rupture. It can be considered to be a preliminary study, and once the present hypothesis has been confirmed, the study can continue looking for a correlation with OA. The evaluations of groups with acute and chronic injury, and groups with or without alteration of muscle mass were added to the manuscript in order to increase the power of the statistical survey. A reflection in this regard has been added to the Discussion section.
Moreover, the conclusion does not constitute serious discovery but rather kind of a wishful thinking. In my opinion this study should be supported by clinical (surgical) trials.
Response 6: I'm sorry you did not appreciate our efforts, but we believe these findings should be pointed out regardless of the surgery. Surely additional studies on the effectiveness of different techniques would be interesting, but it was not the aim of this study. To improve the study, investigations have been added between new groups, differentiated by injury duration and influence of muscle mass. The new results have been inserted into a new table and commented on the Discussion section.
Round 2
Reviewer 1 Report
Thanks for improving the paper.
Reviewer 3 Report
The authors thoroughly rebuilt their article in such a way that it basically gained a new form.
In general, all my comments have been logically explained.
I accept the authors' explanations and corrections and since the article has been significantly improved, I recommend it for publication.